# Apoptotic and DNA Damage Effect of 1,2,3,4,6-Penta-O-galloyl-beta-D-glucose in Cisplatin-Resistant Non-Small Lung Cancer Cells via Phosphorylation of H_2_AX, CHK2 and p53

**DOI:** 10.3390/cells11081343

**Published:** 2022-04-14

**Authors:** Ji-Hyun Kim, Eunji Im, Jihyun Lee, Hyo-Jung Lee, Deok Yong Sim, Ji Eon Park, Chi-Hoon Ahn, Hyeon Hee Kwon, Bum Sang Shim, Bonglee Kim, Sung-Hoon Kim

**Affiliations:** Molecular Cancer Target Herbal Research Laboratory, College of Korean Medicine, Kyung Hee University, Seoul 02447, Korea; kimji77@pusan.ac.kr (J.-H.K.); ji4137@khu.ac.kr (E.I.); jhlee096@naver.com (J.L.); hyonice77@khu.ac.kr (H.-J.L.); simdy0821@khu.ac.kr (D.Y.S.); jieon77@khu.ac.kr (J.E.P.); ach2565@khu.ac.kr (C.-H.A.); dwfqwd223@hanmail.net (H.H.K.); eshimbs@khu.ac.kr (B.S.S.); bongleekim@khu.ac.kr (B.K.)

**Keywords:** PGG, non-small-cell lung cancer (NSCLC), apoptosis, DNA damage, cisplatin

## Abstract

Herein, the apoptotic mechanism of 1,2,3,4,6-penta-O-galloyl-β-D-glucopyranose (PGG) was examined in cisplatin-resistant lung cancer cells. PGG significantly reduced viability; increased sub-G1 accumulation and the number of terminal deoxynucleotidyl transferase (TdT) dUTP Nick-End Labeling (*TUNEL*)-positive cells; induced the cleavage of poly (ADP-ribose) polymerase (PARP), caspases (8,9,3,7), B-cell lymphoma protein 2 (Bcl-2)-associated X (Bax) and phosphatase and tensin homolog deleted on chromosome 10 (PTEN); and attenuated the expression of p-AKT, X-linked inhibitor of apoptosis protein (*XIAP*), Bcl-2, Bcl-xL and survivin in A549/cisplatin-resistant (CR) and H460/CR cells. Notably, PGG activated p53, p-checkpoint kinase 2 (CHK2) and p-H2A histone family member X (p-*H*_2_*AX*), with increased levels of DNA damage (DSBs) evaluated by highly expressed pH2AX and DNA fragmentation registered on comet assay, while p53 knockdown reduced the ability of PGG to reduce viability and cleave caspase 3 and PARP in A549/CR and H460/CR cells. Additionally, PGG treatment suppressed the growth of H460/CR cells in Balb/c athymic nude mice with increased caspase 3 expression compared with the cisplatin group. Overall, PGG induces apoptosis in cisplatin-resistant lung cancer cells via the upregulation of DNA damage proteins such as γ-H_2_AX, pCHK2 and p53.

## 1. Introduction

Lung cancer is the most commonly diagnosed cancer and the leading cause of cancer death among males worldwide according to Global Cancer Statistics 2018 [1]. Among lung cancers, non-small-cell lung cancer (NSCLC), representing approximately 85% of overall lung cancers, still has a poor prognosis, despite the use of tobacco control, chemotherapy, radiotherapy and molecular targeted therapy [2].

Though cisplatin has been used in several solid tumors such as lung, colorectal, testicular and bladder cancers for years as a DNA-damaging anticancer drug, it has some limitations due to drug resistance [3,4]. It has been well documented that cisplatin resistance is induced by cisplatin-elicited signals, the binding of cisplatin to DNA, DNA-cisplatin adducts and cisplatin-mediated DNA damage [5]. Emerging evidence reveals that cisplatin resistance is induced by the activation of phosphatidylinositol-3-kinase/AKT, Wnt/β-catenin and mitogen-activated protein kinase (MAPK) and the inhibition of tumor suppressor genes such as p53 and phosphatase and tensin homolog deleted on chromosome 10 (PTEN) [3,4,6,7], mediated by impaired DNA damage response (DDR) or DNA repair [8]. Furthermore, DDR activation includes the upregulation of ataxia telangiectasia-mutated (ATM) and ataxia telangiectasia and rad3-related (ATR) signaling elicited by the phosphorylation of CHK2, histone H_2_AX and p53, leading to the inhibition of DNA replication and cell-cycle arrest [9,10,11].

1,2,3,4,6-penta-O-galloyl-β-D-glucopyranose (PGG), a water-soluble polyphenolic tannin derived from *Rhus chinensis* or *Terminalia chebula*, was known to have anti-inflammatory [12], anti-oxidant [13], anti-angiogenic [14], anti-metastatic [15], anti-diabetic [16] and apoptotic effects in pancreas [17], colon [18], breast [19], liver [20] and prostate [21] cancers and leukemia [22]. Nevertheless, the apoptotic mechanism of PGG in NSCLCs remains unclear to date. Hence, in the present study, the underlying molecular mechanism of PGG was elucidated in cisplatin-resistant A549 and H460 NSCLCs in association with the DDR signaling pathway.

## 2. Materials and Methods

### 2.1. PGG Preparation

PGG (M.W. = 940.68; CAS No. 14937-32-7) from Sigma Aldrich (Sigma Aldrich, St. Louis, MO, USA) was dissolved in dimethyl sulfoxide (DMSO) and stored as a 100 mM stock for the following experiments.

### 2.2. Cell Culture

A549 and H460 NSCLCs from American Type Culture Collection (ATCC) were maintained in Roswell Park Memorial Institute (*RPMI*) 1640 (Gibco BRL, Grand Island, NY, USA) containing 10% fetal bovine serum (FBS). In addition, A549/CR and H460/CR cells, which were kindly provided by Dr. Jin Kyung Rho, were cultured in RPMI 1640, 10% FBS, 100 units mL^−1^ penicillin/streptomycin and 2 μM of L-glutamine [23].

### 2.3. Cell Viability Assay

The effect of PGG on cell viability was evaluated using a 3-(4,5-dimethylthiazol-2-yl)-2,5-diphenyl tetrazolium bromide (MTT) assay. A549, A549/CR, H460 and H460/CR cells (1 × 10^4^ cells) were exposed to various concentrations of cisplatin (0, 6.25, 12.5, 25 and 50 μM) and/or PGG (0, 6.25, 12.5, 25 and 50 μM). After 24 h or 48 h of culture, the cells were exposed to MTT solution (1mg/mL) for 2 h, and the cell viability was determined as the percentage of viable cells in the PGG-treated group versus the untreated control at optical density (OD) using a microplate reader at 570 nm.

### 2.4. Cell Cycle Analysis

Based on Kwon et al.’s paper [22], A549/CR and H460/CR cells (1 × 10^6^) were exposed to PGG (0, 12.5 and 25 μM) for 48 h, followed by regular cell cycle analysis protocol by staining with propidium iodide (PI, Sigma-Aldrich, St. Louis, MO, USA) (50 μg/mL) for 30 min. The DNA contents of the stained cells were analyzed by FACSCalibur (Becton Dickinson, Franklin Lakes, NJ, USA) using the Cell Quest program (BD Bio-sciences, San Jose, CA, USA).

### 2.5. Annexin-V-FITC Apoptosis Assay

A549, A549/CR, H460 and H460/CR cells after exposure to PGG at a level of 25 μM for 48 h were labeled with an Annexin VFITC/PI apoptosis detection kit (Biovision Inc, Mountain View, CA, USA). Then, the numbers of apoptotic and necrotic cells were determined by the Cell Quest software of BD flow cytometer.

### 2.6. TUNEL Assay

DNA fragments from A549/CR and H460/CR cells exposed to PGG at 25 μM for 48 h were detected using a Dead End^TM^ fluorometric TUNEL assay kit (Promega, Madison, WI, USA). Green fluorescence for apoptotic bodies and red for 4′, 6-diamidino-2-phenylindole (DAPI) detecting nucleus were observed using an Axio vision 4.0 fluorescence microscope (Carl Zeiss Inc., Thornwood, NY, USA).

### 2.7. DNA Fragmentation Assay

Based on Kim et al. [24], DNA cleavages were determined in A549/CR and H460/CR cells exposed to PGG (0, 6.25, 12.5 and 25 μM) for 48 h using agarose gel electrophoresis. In brief, after the supernatant containing the fragmented DNA cells was subjected to lysis and centrifugation, DNA fragmentation was analyzed on 1.5% agarose gels.

### 2.8. Western Blotting and Nuclear Extract Preparation

Based on Lee et al. [25], Western blotting was conducted in A549/CR and H460/CR cells exposed to PGG (0, 6.25, 12.5 and 25 μM) for 48 h. In brief, whole cell lysates were lysed in ice-cold radioimmunoprecipitation assay (RIPA) buffer containing protease inhibitor cocktail, and isolated proteins in the supernatants were quantified and electrotransferred onto a Hybond enhanced chemiluminescence (ECL) transfer membrane. The membranes were probed with primary antibodies for PTEN, p-AKT, PARP, cellular inhibitor of apoptosis 1 (c-IAP1), c-IAP2, Survivin (Santa Cruz Biotechnologies, Santa Cruz, CA, USA), Caspase-8, -9, -7, Cleaved caspase-3, BAX, Bcl-2, Bcl-xL, p-ATR, p-Chk2, p-BRCA-1, p-H_2_AX (Cell signaling Technology, Danvers, MA, USA), p53 (Oncogene Research Products, San Diego, CA, USA), XIAP (Becton Dickinson and Company BD Biosciences, San Jose, CA, USA), β-actin (Sigma Aldrich, St Louis, MO) and horseradish peroxidase-conjugated secondary antibody. Additionally, nuclear extract was isolated for DNA damage proteins using an NE-PER Nuclear Cytoplasmic Extraction Reagent kit (Thermo Scientific, Rochester, NY, USA).

### 2.9. siRNA Transfection and p53 Luciferase Assay

A549/CR and H460/CR cells were transfected with control or p53 siRNAs using Interferin^TM^ transfection reagent (Polyplus-transfection Inc., New York, NY, USA). Luciferase activity was determined using a p53 luciferase reporter assay kit and a LUMIstar OPTIMA luminometer plate reader (BMG Labtech, Offenburg, Germany) after A549/CR and H460/CR cells were transiently transfected with p53 reporter gene and exposed to PGG (12.5 and 25 μM) for 48 h.

### 2.10. Immunofluorescence Assay

Based on Jung et al. [26], A549/CR and H460/CR cells treated with PGG (12.5 and 25 μM) for 48 h were plated onto a Lab-Tek II Chamber slide (Thermo Scientific, Rochester, NY, USA) and exposed to p-H_2_AX as a primary antibody (Cell signaling Technology, Danvers, MA, USA), then exposed to Alexa Fluor 546-conjugated secondary antibody (Invitrogen, Waltham, MA, USA). Then, the stained cells were visualized in DAPI-containing medium using an Olympus FLUOVIEW FV10i (Olympus, Tokyo, Japan) Confocal Microscope (×60).

### 2.11. Comet Assay

A549/CR and H460/CR cells treated with PGG (12.5 and 25 μM) for 48 h were evaluated for DNA damage using a Trevigen Comet Assay™ kit (Trevigen Inc., Gaithersburg, MD, USA). The cells were subjected to regular Comet assay, and the isolated DNAs were stained with SYBR Green I dye (Trevigen, 1:10,000 in Tris–EDTA buffer, pH 7.5) for 20 min and visualized using an Axio vision 4.0 fluorescence microscope (Carl Zeiss Inc., Dublin, CA, USA).

### 2.12. Combination Index

Combination index (CI) was evaluated using the Chou–Talalay method and CalcuSyn software (Biosoft, Ferhuson, MO, USA). A CI of less than 1 was determined to be synergistic [24,27].

### 2.13. Tumor Xenograft Model and Immunohistochemistry

This animal experiment was approved by Kyung Hee University’s Institutional Animal Care and Use Committee (Approval No. KHUASP(SE)-11-001). BALB/c athymic nude mice (6 weeks old, male) purchased from Jung Ang lab animal (Seoul, Korea) adapted to the classical condition for a week. The mice were divided into three groups (*n* = 4 per group): vehicle control (1% DMSO with saline), PGG (2 mg/kg body weight) and cisplatin (2 mg/kg body weight). Then, H460/CR cells (2.5 × 10^6^) mixed with Matrigel (Becton Dickinson, Franklin Lakes, NJ, USA) were subcutaneously injected into the right flank of the mice. One week after implantation, the mice were orally administered PGG or intraperitonially (i.p) injected with cisplatin daily for 12 days. Tumor volume was monitored every other day with caliper, and immunohistochemistry was conducted for caspase 3 based on Jung et al. [28].

### 2.14. Statistical Analyses

The data represent the means ± SD. The statistical significance was analyzed between two groups with Student’s unpaired *t*-test, while the significance among multiple groups (more than two) was determined using a Tukey–Kramer post hoc test after ANOVA.

## 3. Results

### 3.1. PGG Reduces the Viability of Human Cisplatin-Resistant Lung Cancer Cells Compared with Their Parental Cells

Previous evidence reveals that the overexpression of tumor suppressor PTEN reduces multidrug resistance (MDR) in chemotherapy, including cisplatin [5]. To verify the resistance of A549/CR and H460/CR cells, the effect of cisplatin or PGG on viability (Figure 1A) and the PTEN expression level was examined in A549 and H460 parental cells and cisplatin-resistant A549/CR and H460/CR cells using an MTT assay and Western blotting. Cisplatin-resistant cell lines demonstrated a subtle reduction in viability upon treatment with increasing doses of cisplatin compared with the parental cell line. In addition, PTEN expression was lower in the A549/CR and H460/CR cells than in the parental cells (Figure 1B), indicating the possibility of PTEN-dependent cytotoxicity based on previous evidence that PTEN overexpression inhibits the proliferation of A549 cells, while PTEN depletion exerts the opposite effect [29,30]. Of note, PGG more significantly reduced the viability of the A549/CR and H640/CR cells in a concentration- and time-dependent manner compared with that of the A549 and H460 cells (Figure 1C,D).

### 3.2. PGG Induces Apoptosis in A549/CR and H460/CR Cells

To investigate whether the cytotoxicity of PGG is associated with apoptosis in A549/CR and H460/CR cells, an Annexin V-FITC/PI apoptosis assay, cell cycle analysis and TUNEL assay were conducted in PGG-treated A549/CR and H460/CR cells. PGG increased apoptotic cells to levels of 11.47% and 12.95%, respectively, in the A549/CR and H460/CR cells, both more than in the A549 (3.5%) and H460 (7.8%) cells by Annexin V-FITC/PI staining (Figure 2A). PGG also increased the sub-G1 populations to 29.43% and 25.64%, respectively, in the A549/CR and H460/CR cells, in contrast with the untreated control (Figure 2B). Consistently, the TUNEL assay revealed that PGG increased the number of TUNEL-positive cells in PGG-treated A549/CR and H460/CR cells compared with the untreated control (Figure 2C).

### 3.3. PGG Induces DNA Fragmentation and Affects Apoptosis Related Proteins in A549/CR and H640/CR Cells

PGG induced DNA fragmentation in a concentration-dependent fashion in A549/CR and H640/CR cells (Figure 3A). PGG also cleaved caspase-7, -8, -9 and -3 and PARP and enhanced the expression of BAX in A549/CR and H460/CR cells (Figure 3B and Appendix A). In contrast, PGG decreased the expression of anti-apoptotic proteins such as XIAP, c-IAP1, c-IAP2, Bcl-xL and survivin. However, the expression of XIAP was decreased in PGG-treated H460/CR cells in a concentration-dependent fashion, but not in A549/CR cells (Figure 3B and Appendix A). To validate caspase-dependent apoptosis, an inhibitor study was conducted in the A549/CR and H640/CR cells. Herein, the pancaspase inhibitor Z-VAD-FMK significantly inhibited the ability of PGG to attenuate the expression of procaspase 7, 3, and proPARP in A549/CR and H640/CR cells (Figure 3C).

### 3.4. PGG Induces DNA Damage Proteins in A549/CR and H460/CR Cells

To check the role of DNA damage in PGG-induced apoptosis in A549/CR and H460/CR cells, the effect of PGG on DNA damage proteins in A549/CR and H460/CR cells was evaluated. As shown in Figure 4A, Comet assay revealed that PGG increased bright green comet tails for DNA damage in A549/CR and H460/CR cells. Consistently, immunofluorescence showed that PGG increased the expression of pH_2_AX in A549/CR and H460/CR cells (Figure 4B), since pH_2_AX is required for apoptosis in DNA-damaged cells [31]. Furthermore, as shown in Figure 4C, PGG significantly induced the phosphorylation of H_2_AX, CHK2 and p53 for DNA damage and decreased the phosphorylation of p-ATR and p-BRCA-1 in A549/CR and H460/CR cells (Appendix A), implying that PGG induces DNA damage. Interestingly, PGG induced the phosphorylation of p53 at s20 in the nuclei of A549/CR and H460/CR cells.

### 3.5. PGG Induced PTEN and Reduced p-AKT in A549/CR and H460/CR Cells

Since tumor suppressor PTEN is a known PI3K/AKT inhibitor, a loss of PTEN is involved in cancer progression and resistance to anticancer drugs in several types of cancers [32]. Herein, PGG induced the expression of PTEN and reduced the phosphorylation of AKT in a concentration-dependent fashion in A549/CR and H460/CR cells (Figure 5A and Appendix A). However, PGG attenuated the expression of even the level of endogenous AKT (Figure 5B and Appendix A), implying some toxicity, which should be explored in the future.

### 3.6. The Pivotal Role of p53 in PGG Induced Apoptosis in A549/CR and H460/CR Cells

It has been well documented that p53 is critically associated with DNA damage response, as the DNA damage-responsive kinases such as ATM and ATR and their downstream mediators CHK1 and CHK2 phosphorylate p53 at Ser15 and Ser20 [33,34], leading to p53-dependent apoptosis in several cancers [35]. Here, PGG increased the protein expression of p53 (Figure 6A and Appendix A) and its luciferase activity (Figure 6B) in a concentration-dependent fashion in A549/CR and H460/CR cells. Conversely, p53 depletion hindered the ability of PGG to reduce the viability (Figure 6C) and to cleave caspase 3 and PARP in A549/CR and H460/CR cells (Figure 6D and Appendix A).

### 3.7. PGG Reduces the Growth of H460/CR Cells Implanted on the Right Flank of Balb/c Nude Mice with Increased Expression of Caspase 3 Compared to Cisplatin Treated Control Group

To validate the in vitro apoptotic effect of PGG, an animal study was conducted in a H460/CR xenograft model. After the treatment of PGG or cisplatin for 12 days following H460/CR cell injection into the flank of Balb/c nude mice (Figure 7A), PGG treatment showed no toxicity and weight loss (Figure 7B), and it also suppressed the tumor sizes with an increased expression of caspase 3 (Figure 7C–E).

## 4. Discussion

In the current study, the underlying apoptotic mechanism of a hydrophilic tannin PGG was explored in association with DNA damage response (DDR) signaling in cisplatin-resistant A549 and H460 lung cancer cells. Here, PGG significantly reduced viability and increased the subG1 accumulation and the number of TUNEL-positive cells in A549/CR and H460/CR cells compared with parent lung cancer cells, indicating the apoptotic potential of PGG to overcome chemoresistance to cisplatin.

Emerging evidence reveals that cisplatin resistance is critically associated with increased DNA repair, DNA damage tolerance, inactivation of caspases, pro-apoptotic proteins such as BAX or Bad and loss of tumor suppressor p53 and PTEN [3,5]. Generally, apoptosis is induced mainly by the intrinsic pathway for caspase-9/3 activation or the extrinsic or death receptor pathway for caspase-8/3 or 7 [36]. It has been well documented that chemoresistance is induced by the inhibition of pro-apoptotic proteins such as PTEN, BAX and Bad and the overexpression of anti-apoptotic proteins such as p-AKT, survivin, XIAP, Bcl-2 and Bcl-xL in resistant cancer cells [37,38]. Herein, PGG enhanced the cleavages of PARP, caspases (8,9,3,7), PTEN and BAX and also reduced the expression of antiapoptotic proteins such as p-AKT, XIAP, Bcl-2, Bcl-xL and survivin in A549/CR and H460/CR cells compared with intact parental cells. In addition, pancaspase inhibitor Z-VAD-FMK significantly disturbed the ability of PGG to attenuate the expression of procaspase 7 and 3 and proPARP in A549/CR and H640-CR cells, indicating that the caspase-dependent apoptotic effect of PGG is mediated by the inhibition of anti-apoptotic proteins in A549/CR and H460/CR cells.

DDR is known to detect damage and coordinate multiple pathways for cell cycle arrest or apoptosis induction, given that the inactivation of DDR mediates tumor progression [5]. Among DDR proteins, including ATM, ATR and DNA-dependent protein kinase (DNA-PK), modulate the initial sensing of double-strand break (DSB), leading to the phosphorylation of p53 [39], while Chk-2 and Chk-1 are activated by ATM and ATR, respectively [40]. Additionally, accumulating evidence reveals that p-ATR is overexpressed in several cancers [41,42,43]. During the DDR process, double strand break (DSB) induces the phosphorylation of H2AX (γ-H2AX). Notably, PGG activates p53, p-CHK2 and p-H_2_AX, decreases p-ATR as a potent ATR inhibitor along with DNA fragmentation, and increases the number of Comet tails and fluorescent p-H2AX by immunofluorescence in A549/CR and H460/CR cells; findings demonstrated that DDR proteins such as p53, p-CHK2 and p-H2AX are critically involved in PGG-induced apoptosis in A549/CR and H460/CR cells. Interestingly, PGG activated p53 protein and its luciferase activity, while p53 depletion suppressed the ability of PGG to reduce viability and induce cleavages of caspase 3 and PARP in A549/CR and H460/CR cells, implying the important role of p53 in PGG-induced apoptosis in A549/CR and H460/CR cells.

Interestingly, previous evidence reveals that the combinatorial treatment of single compound and cisplatin enhances apoptosis in cisplatin-resistant cancer cells such as human oral cancer [44], colorectal cancer [37] and ovarian cancer [45]. Here, the combined treatment of PGG and cisplatin significantly reduced viability and increased sub-G1 population compared with PGG or cisplatin alone in A549/CR cells and H460/CR cells. Consistently, combined treatment of PGG and cisplatin enhanced the cleavages of caspases (8,9,7,3) and PARP and also attenuated the expression of XIAP, c-IAP and Bcl-2 compared with PGG or cisplatin alone in A549/CR and H460/CR cells (Appendix A). Nonetheless, to confirm the above in vitro results, an animal study with optimal doses of cisplatin and PGG is required.

Furthermore, PGG suppressed the growth of H460/CR cells in Balb/c nude mice with an increased expression of caspase 3 compared with the cisplatin-treated group, demonstrating that PGG exerts an apoptotic effect in a cancer xenograft model consistent with its in vitro effect.

In summary, PGG significantly reduced viability; increased sub-G1 accumulation and TUNEL-positive cells; induced the cleavages of PARP, caspases (8,9,3,7), PTEN, BAX, p53, p-CHK2 and p-H_2_AX; increased DNA fragmentation, Comet tails and fluorescent p-H_2_AX; and attenuated the expression of p-AKT, XIAP, Bcl-xL and survivin in A549/CR and H460/CR cells. Conversely, p53 depletion reduced the ability of PGG to cleave caspase 3 and PARP in A549/CR and H460/CR cells. Furthermore, the co-treatment of PGG and cisplatin synergistically reduced viability; increased sub-G1 accumulation and cleavages of PARP and caspases (8,9,3,7); and attenuated the expression of c-IAP and Bcl-2 in A549/CR and H460/CR cells. Additionally, PGG suppressed the growth of H460/CR cells in Balb/c nude mice with an increased expression of caspase 3. Overall, these findings provide evidence that PGG induces apoptosis in cisplatin-resistant NSCLCs via the upregulation of DNA damage proteins as γ-H_2_AX, pCHK2 and p53 and inhibition of anti-apoptotic proteins as a potent anticancer agent for cisplatin-resistant NSCLC treatment (Figure 8).

## 5. Conclusions

Through in vitro and in vivo studies to explore the apoptotic mechanism and combinatorial potential of PGG with cisplatin in cisplatin-resistant lung cancer cells, it can be concluded that PGG induces apoptosis in cisplatin-resistant NSCLCs via the upregulation of DNA damage proteins such as γ-H_2_AX, pCHK2 and p53 and the inhibition of anti-apoptotic proteins through combination with cisplatin in cisplatin-resistant NSCLCs.

## Figures and Tables

**Figure 1 cells-11-01343-f001:**
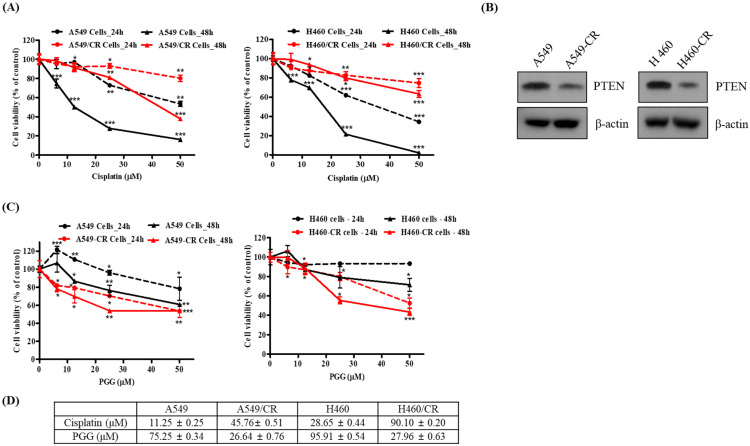
Effect of PGG on the viability and different endogenous PTEN levels in A549/A549CR and H460/H460CR cells. (**A**) Cisplatin-resistant NSCLC lines demonstrated an increased sensitivity to PGG in a cell viability test (MTT). Plots representing cell viability in A549 (shown in black) and A549/CR (shown in red) lines treated with cisplatin (at concentrations of 0, 6.25, 12.5, 25 and 50 μM) for 24 (dashed line) and 48 h (non-dashed line). *, *p* < 0.05, **, *p* < 0.01, ***, *p* < 0.001 vs. untreated control. (**B**) Different endogenous PTEN levels in A549/A549CR and H460/H460CR cells. (**C**) Effect of PGG on the viability of A549/A549CR and H460/H460CR cells. Three independent experiments were conducted. (**D**) IC50 values of cisplatin or PGG in A549/A549CR and H460/H460CR cells. Data represent means ± S.D.

**Figure 2 cells-11-01343-f002:**
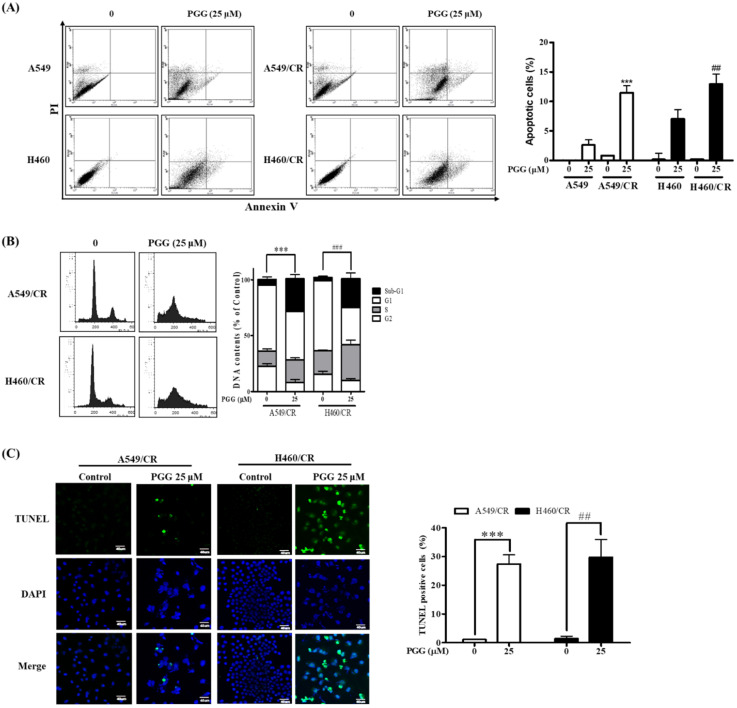
Effect of PGG on apoptosis in A549/CR and H460/CR cells. (**A**) Effect of PGG on apoptosis in A549, A549/CR, H460 and H460/CR cells by Annexin V/PI staining. The lower right quadrant shows Annexin V^+^ PI^−^ cells in early apoptosis, while the upper right quadrant represents Annexin V^+^ PI^+^ cells in late apoptosis. Bar graphs represent the percentages of DNA contents undergoing apoptosis. Data represent means ± S.D. ***, *p* < 0.001 vs. untreated control. (**B**) Effect of PGG on apoptosis in A549/CR and H460/CR cells by cell cycle analysis. Bar graphs represent the percentages of DNA contents undergoing apoptosis. Data represent means ± S.D. ***, *p* < 0.001 vs. untreated control in A549CR cells. ###, *p* < 0.001 vs. untreated control in H460CR cells. (**C**) Effect of PGG on TUNEL-positive cells in A549/CR and H460/CR cells. Arrows indicate TUNEL-stained cells (×60). Bar graphs represent the percentages of TUNEL-positive cells. ***, *p* < 0.001 vs. untreated control, ##, *p* < 0.01 vs. untreated control. Three independent assays were conducted.

**Figure 3 cells-11-01343-f003:**
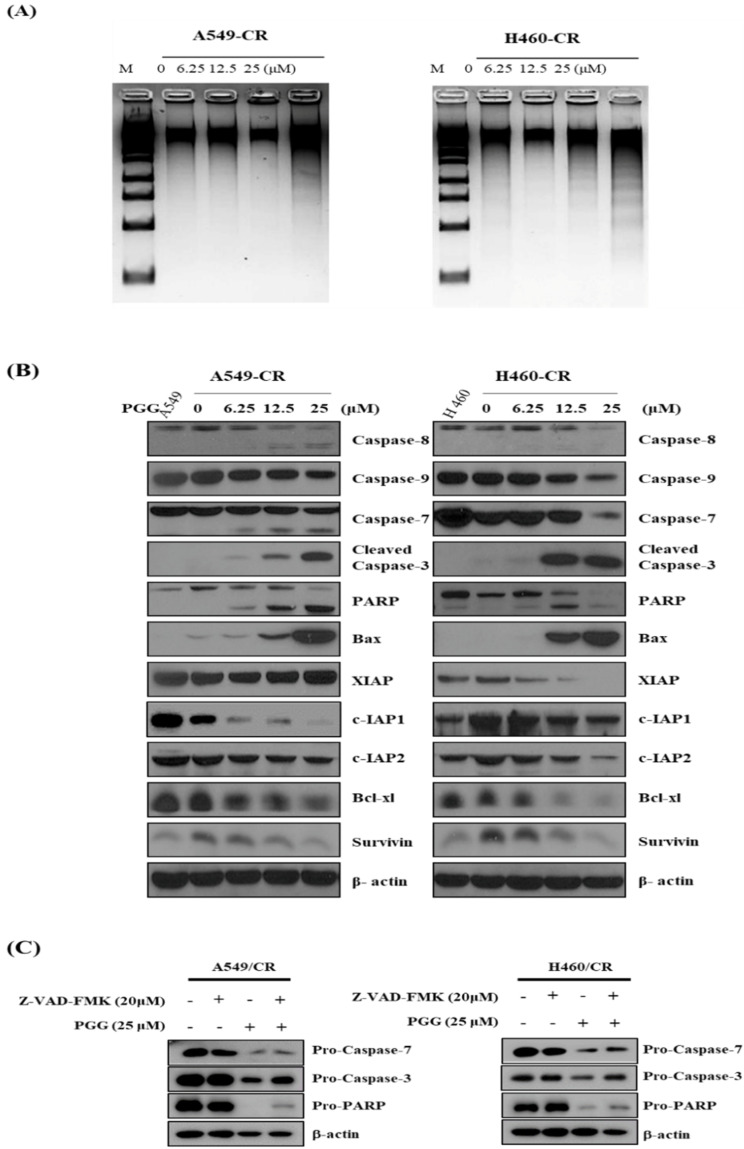
Effect of PGG on DNA fragmentation and apoptosis-related proteins in A549/CR and H460/CR cells. (**A**) Effect of PGG on DNA fragmentation in A549/CR and H460/CR cells. (**B**) Effect of PGG on apoptosis-related proteins in A549/CR and H460/CR cells. (**C**) Effect of pancaspase inhibitor Z-VAD-FMK on apoptosis-related proteins in A549/CR and H460/CR cells. Three independent experiments were conducted.

**Figure 4 cells-11-01343-f004:**
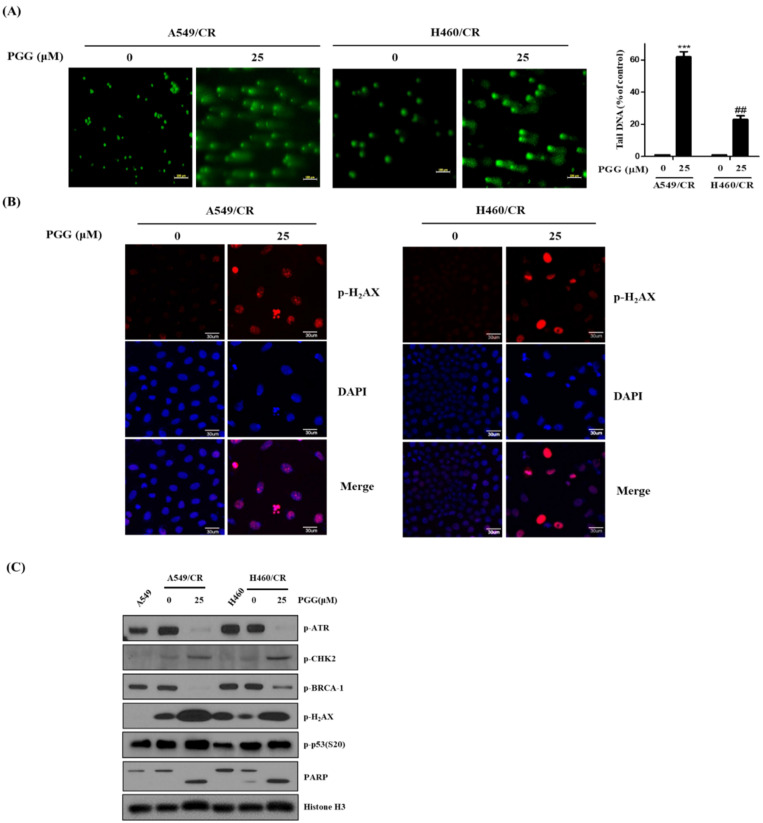
Effect of PGG on DNA damage proteins in A549/CR and H460/CR cells. (**A**) Effect of PGG on the comet tails in A549/CR and H460/CR cells by Comet assay. ***, *p* < 0.001 vs. untreated control, ##, *p* < 0.01 vs. untreated control. (**B**) Effects of PGG on pH_2_AX in A549/CR and H460/CR cells by immunofluorescence (×60). (**C**) Effects of PGG on DNA damage proteins in A549/CR and H460/CR cells. Three independent assays were conducted.

**Figure 5 cells-11-01343-f005:**
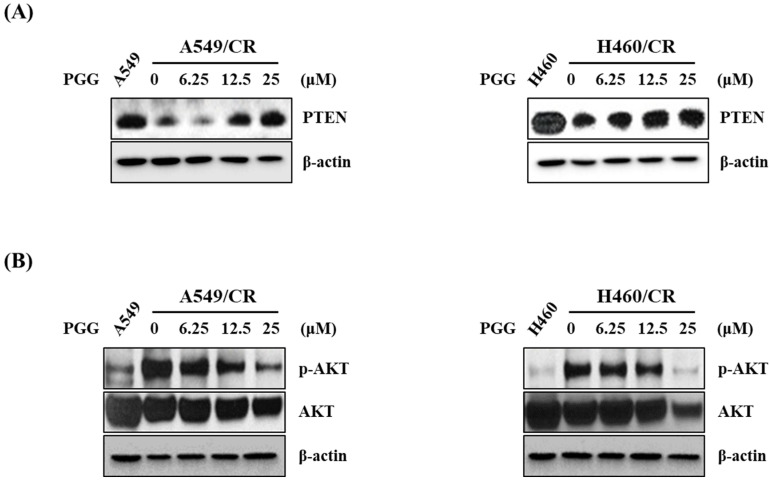
Effect of PGG on the expression of PTEN and p-AKT in A549/CR and H460/CR cells. (**A**) Effect of PGG on PTEN in A549/CR and H460/CR cells (**B**) Effect of PGG on p-AKT in A549/CR and H460/CR cells.

**Figure 6 cells-11-01343-f006:**
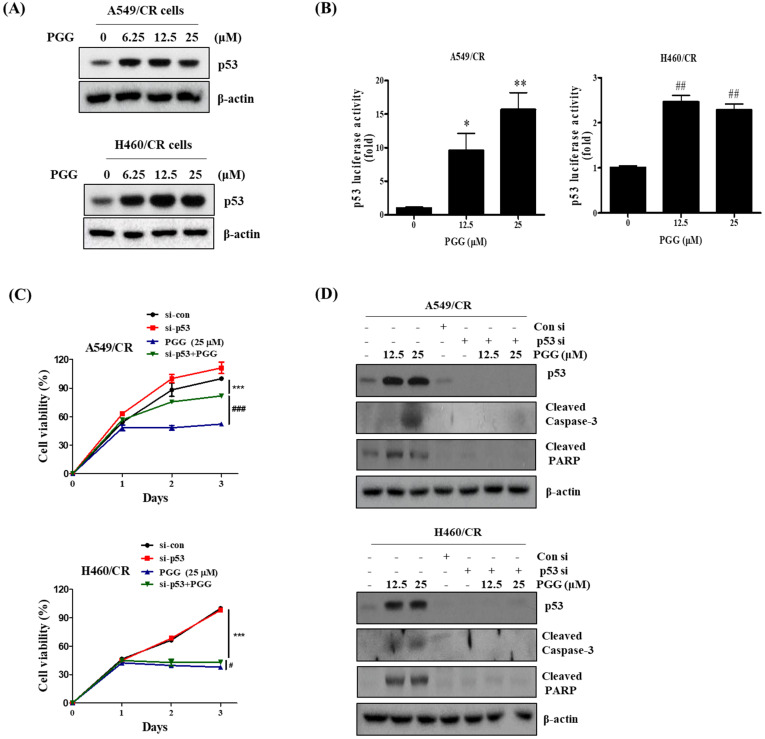
The pivotal role of p53 in PGG-induced apoptosis in A549/CR and H460/CR cells. (**A**) Effect of PGG on the protein expression of p53 in A549/CR and H460/CR cells. *, *p* < 0.05, **, *p* < 0.01 vs. untreated control. ##, *p* < 0.01 vs. untreated control. (**B**) Effect of PGG on p53 luciferase activity in A549/CR and H460/CR cells. (**C**) Effect of p53 depletion on the viability of A549/CR and H460/CR cells. ***, *p* < 0.001 vs. untreated control. #, *p* < 0.05, ###, *p* < 0.001 vs. untreated control. (**D**) Effect of p53 depletion on PARP and cleaved caspase 3 induced by PGG in A549/CR and H460/CR cells. Three independent assays were conducted.

**Figure 7 cells-11-01343-f007:**
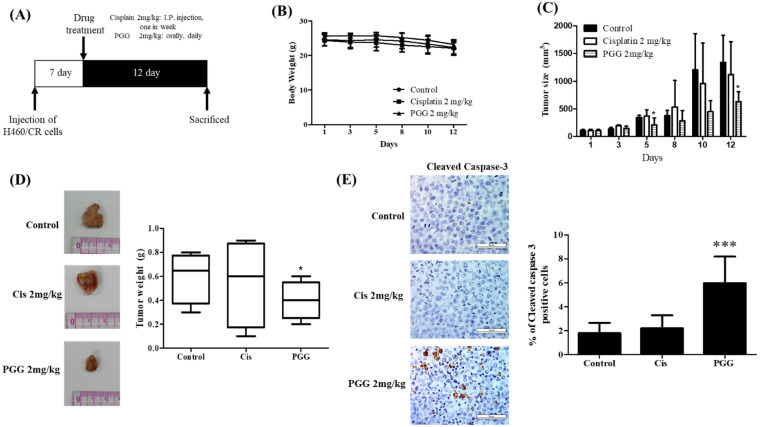
PGG suppresses the growth of H460/CR cells in BALB/c nude mice with increased caspase 3. (**A**) Time schedule of animal study with PGG or cisplatin at a dose of 2mg/kg. (**B**) Effect of PGG treatment on body weight. (**C**) Effect of PGG treatment on tumor volume in mice implanted with H460/CR cells. (**D**) Tumor morphology and tumor weight of mice treated by PGG or cisplatin. *, *p* < 0.05 vs. cisplatin control (**E**) Effect of PGG or cisplatin on cleaved caspase 3 in tumor sections isolated from mice. Immunostaining of cleave caspase-3 (scale bar = 50 μm). Cleaved caspase-3-positive cells were counted in 10 random areas at 400× magnification and analyzed using image J. Quantification represents the percentage of cleaved caspase 3 positive cells. Data represent means ± SD. *** *p* < 0.001 vs. untreated control.

**Figure 8 cells-11-01343-f008:**
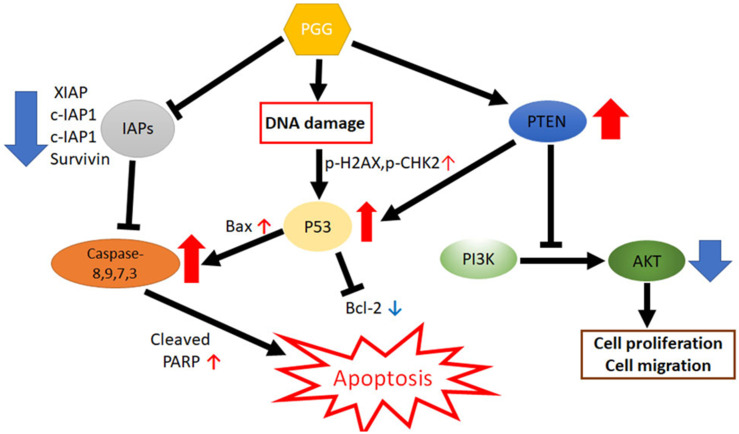
Molecular mechanisms of PGG induced apoptosis via the activation of DNA damage proteins such as p-CHK2, p-γH2AX and p53.

## Data Availability

Not applicable.

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
