# Peer review of "Apoptotic and DNA Damage Effect of 1,2,3,4,6-Penta-O-galloyl-beta-D-glucose in Cisplatin-Resistant Non-Small Lung Cancer Cells via Phosphorylation of H_2_AX, CHK2 and p53"

_cells, 2022, doi:10.3390/cells11081343_

Round 1

Reviewer 1 Report

As I have already stated in previous reviews, the data presented in the study represent interesting and valuable information regarding the molecular mechanism of PGG  in cisplatin resistant A549 and H460 non-small cell lung cancer. Since in the new versions of the manuscript authors responded to the concerns made in previous revisions and improved its content and quality, I believe that in the current form presented results are valuable observations and the study is of sufficient significance and meets the standards required for publishing in the Cells.

Author Response

We appreciate you so much for positive comments.

Reviewer 2 Report

Dear Authors,

I’m writing here to submit a revision for an article that I have recently received. The article “Apoptotic and DNA Damage Effect of 1,2,3,4,6-Penta-O-2 Galloyl-Beta-D-Glucose in Cisplatin Resistant Non-Small Lung Cancer Cells via Phosphorylation of γ-H2AX, CHK2 and p53 ” by J.-H. Kim et al., aimed on characterization of natural compound 1, 2, 3, 4, 6-penta-O-galloyl-β-D-glucopyranose (PGG) as a potential novel tool for treatment of a group of solid tumors – cisplatin resistant Non-Small Lung  Cancers. The authors propose that anticancer activity of 1, 2, 3, 4, 6-penta-O-galloyl-β-D-glucopyranose (PGG) is due to inhibition of tumor growth by inducing apoptosis. The current work was performed using cell lines, representing the common types of NSCLC lines: A549 and H460, as well as cisplatin resistant lines A549/CR and H460/CR.  Experiments and assays used in the current study are up-to-date and well established. The study is not limited by in-vitro experiments and supplemented by in-vivo experiments (xenograft tumor model), that definitely have significant value for the current study.

The work is  intriguing and has serious problems with data presentation and it requires a lot of effort to bring it to the shape that could fit the standards for Cells journal. In my understanding some experiments are misleading and at times impossible to interpret probably because figures were not sorted for main and supplementary folders, I would strongly recommend moving some figures in the supplementary folder . 

Nevertheless, here is the list of my main concerns and comments about the current paper:

1) Figure 1C. The authors claim that PGG significantly reduced the viability of A549/CR and H640/CR cells in a concentration- and time-dependent manner more than A549 and H460 cells. 

  1. As a result of observation could it be that PGG sensitizes CR lines to cisplatin. It would be a valuable addition to show the synergic effect could have a place that will dramatically raise the value of information presented here, to see if even low concentrations of PGG re-sensitize resistant lines to cisplatin. 
  2. As a result of observation WB data of PTEN expression is shown. Please describe in the text this finding and prove suggested phenomenon with siRNA to PTEN treatment of parental lines, do they have similar response to PGG? 

2) I would suggest analyzing the most relevant experiments side-by-side with cisplatin-sensitive lines. At least for  FACS, TUNEL and IF experiments to demonstrate that effect is more robust in Cis-resistant lines, this is in line with idea that the authors proposed.

3) I have concerns about Figure 2 A and B. The data presented in figure 2A is aimed to demonstrate apoptotic events in cisplatin resistant lines upon treatment with 25uM PGG compared to control groups (untreated cells). Unfortunately, according to the data presented on figure 2A is poor resolution and quality. There is the impression that the majority of events are below the X-axis and very few of Annexin positive cells (right quadrants) are visible in the field, which is much less than 15% indicated in the plot. I would suggest adding samples with Cisplatin (25uM) on Cisplatin-resistant and parental lines here (because it was indicated in the Materials and Methods section but I could not find that data in the article). And using 48h 25uM Cisplatin treated parental lines as positive control to indicate annexinV positive events for all FACS experiments because according to the plots on Figure 1A this condition causes dramatic reduction in viability. 

Ref for Annexin V/PI staining. Mendonca, P., Alghamdi, S., Messeha, S. et al. Pentagalloyl glucose inhibits TNFαactivated CXCL1/GRO-α expression and induces apoptosisrelated genes in triple-negative breast cancer cells. Sci Rep 11, 5649 (2021). https://doi.org/10.1038/s41598-021-85090-z

4) Figure 4B. I would suggest changing the "-" sign to "0" ( for images with control samples) and increase DAPI intensity in H460/CR line images .

5)     Xenograft animal model. Unfortunately there is no statistical analysis or statistical data is not presented of figures characterizing tumor size and weight.  It is difficult to fix but there is no data about combinational effect Cis+PGG.

Minor comments about design and grammar.

  1. Scale bar is missing or barely visible: Figure 2C; Figure 4A,B; Figure 7E; 
  2. Fonts are not consistent among Figures. Please use one standard font for all figures.
  3. WB data presentation: please be consistent using /not using frames for WB data.
  4. Please mark each plot or table in figures with separate letters and individual legend. For example, the table in the figure does not contain letters. Figure legends should be improved and more informative than "effect of PGG on" . For example: Figure 1. Cisplatin resistant NSCLC lines demonstrate increased sensitivity to PGG in cell viability tests (MTT). A. Plot representing cell viability in A549 (shown in black) and A549/CR (shown in red) lines treated with Cisplatin (at concentrations 0, 6.25, 12.5, 25 and 50 μM) for 24 (dashed line) and 48 hours (non-dashed line)... And so on.
  5. Figures 3,4,5,6 contain densitometry plots that should be moven in the supplementary folder.
  6. Statistical difference or NS should be added at least for figures: 1A; 1C; 7B; 7D.
  7. Changes in title Line 4: "…Phosphorylation of γ-H2AX…" to "…Phosphorylation of H2AX…", because γ-H2AX is already a phosphorylated form (Ser139) of H2AX.
  8. Please be consistent with abbreviations in the text (for example using γ-H2AX or pH2AX (S139); Table in figure 1A (please indicate that IC50 value was calculated for 48htimepoint  otherwise calculate GI50). 
  9. Comet assay should be quantified according to available protocols and data presented as plots along with representative images in figure 4A. 
  10. Line 164: "Cisplatin did not reduce the viability of A549/CR and H640/CR cells compared to parental cells in a time and concentration dependent fashion" change to "Cisplatin resistant cell lines demonstrated subtle reduction in viability upon treatment with increasing doses of cisplatin compare to parental cell lines (Figure 1A)".  
  11. Line 166: referred figure is not correct: Figure 1B.
  12.  Labels are not aligned (for example figure 3A, 4C "parental lines"; 5A middle WB; )

Author Response

The work is intriguing and has serious problems with data presentation and it requires a lot of effort to bring it to the shape that could fit the standards for Cells journal. In my understanding some experiments are misleading and at times impossible to interpret probably because figures were not sorted for main and supplementary folders, I would strongly recommend moving some figures in the supplementary folder . 

Nevertheless, here is the list of my main concerns and comments about the current paper:

1) Figure 1C. The authors claim that PGG significantly reduced the viability of A549/CR and H640/CR cells in a concentration- and time-dependent manner more than A549 and H460 cells. 

  1. As a result of observation could it be that PGG sensitizes CR lines to cisplatin. It would be a valuable addition to show the synergic effect could have a place that will dramatically raise the value of information presented here, to see if even low concentrations of PGG re-sensitize resistant lines to cisplatin. 
  2. As a result of observation WB data of PTEN expression is shown. Please describe in the text this finding and prove suggested phenomenon with siRNA to PTEN treatment of parental lines, do they have similar response to PGG? 

Response) Thanks. We described this finding in the text by citation of previous reference that PTEN overexpression decreased proliferation of A549 cells, while PTEN depletion showed the opposite effect (Biomed Res Int. 2016;2016:2476842. doi: 10.1155/2016/2476842. Epub 2016 Oct 16.PTEN Inhibits Cell Proliferation, Promotes Cell Apoptosis, and Induces Cell Cycle Arrest via Downregulating the PI3K/AKT/ hTERT Pathway in Lung Adenocarcinoma A549 Cells)

2) I would suggest analyzing the most relevant experiments side-by-side with cisplatin-sensitive lines. At least for FACS, TUNEL and IF experiments to demonstrate that effect is more robust in Cis-resistant lines, this is in line with idea that the authors proposed.

Response) Thanks for your comments. We confirmed that PGG increased the number Annexin V apoptotic cells in A549/CR and H640/CR cells more than in A549 and H460 cells by further experiment by using Annexin VFITC/PI apoptosis detection kit as shown in Figure 2A.

3) I have concerns about Figure 2 A and B. The data presented in figure 2A is aimed to demonstrate apoptotic events in cisplatin resistant lines upon treatment with 25uM PGG compared to control groups (untreated cells). Unfortunately, according to the data presented on figure 2A is poor resolution and quality. There is the impression that the majority of events are below the X-axis and very few of Annexin positive cells (right quadrants) are visible in the field, which is much less than 15% indicated in the plot. I would suggest adding samples with Cisplatin (25uM) on Cisplatin-resistant and parental lines here (because it was indicated in the Materials and Methods section but I could not find that data in the article). And using 48h 25uM Cisplatin treated parental lines as positive control to indicate annexinV positive events for all FACS experiments because according to the plots on Figure 1A this condition causes dramatic reduction in viability. 

Ref for Annexin V/PI staining. Mendonca, P., Alghamdi, S., Messeha, S. et al. Pentagalloyl glucose inhibits TNF‐α‐activated CXCL1/GRO-α expression and induces apoptosis‐related genes in triple-negative breast cancer cells. Sci Rep 11, 5649 (2021). https://doi.org/10.1038/s41598-021-85090-z

Response) Thanks. New data were added in Figure 2A based on your comments.

4) Figure 4B. I would suggest changing the "-" sign to "0" ( for images with control samples) and increase DAPI intensity in H460/CR line images .

Response) Thanks. Corrected as “0”

5)     Xenograft animal model. Unfortunately there is no statistical analysis or statistical data is not presented of figures characterizing tumor size and weight.  It is difficult to fix but there is no data about combinational effect Cis+PGG.

Response) Thanks. Statistical significance was added. But we mentioned the necessity of further animal study with optimal doses of Cis+PGG in Discussion.

Minor comments about design and grammar.

  1. Scale bar is missing or barely visible: Figure 2C; Figure 4A,B; Figure 7E; 

Response) Corrected

  1. Fonts are not consistent among Figures. Please use one standard font for all figures.

Response) Thanks. Corrected

  1. WB data presentation: please be consistent using /not using frames for WB data.

Response) Thanks. Corrected

  1. Please mark each plot or table in figures with separate letters and individual legend. For example, the table in the figure does not contain letters. Figure legends should be improved and more informative than "effect of PGG on" . For example: Figure 1. Cisplatin resistant NSCLC lines demonstrate increased sensitivity to PGG in cell viability tests (MTT). A. Plot representing cell viability in A549 (shown in black) and A549/CR (shown in red) lines treated with Cisplatin (at concentrations 0, 6.25, 12.5, 25 and 50 μM) for 24 (dashed line) and 48 hours (non-dashed line)... And so on.

Response) Thanks for your kind comments. We corrected these figures based on your comments.

  1. Figures 3,4,5,6 contain densitometry plots that should be moven in the supplementary folder.

Response)Thanks. Densitometry plot data were moved in the supplementary folder.

  1. Statistical difference or NS should be added at least for figures: 1A; 1C; 7B; 7D.

Response)Thanks. Added

  1. Changes in title Line 4: "…Phosphorylation of γ-H2AX…" to "…Phosphorylation of H2AX…", because γ-H2AX is already a phosphorylated form (Ser139) of H2AX.

Response) Corrected.

  1. Please be consistent with abbreviations in the text (for example using γ-H2AX or pH2AX (S139); Table in figure 1A (please indicate that IC50 value was calculated for 48h timepoint  otherwise calculate GI50). 

Response) Thanks. IC50 Table was added with legend

  1. Comet assay should be quantified according to available protocols and data presented as plots along with representative images in figure 4A. 

Response) Thanks. Quantified

  1. Line 164: "Cisplatin did not reduce the viability of A549/CR and H640/CR cells compared to parental cells in a time and concentration dependent fashion" change to "Cisplatin resistant cell lines demonstrated subtle reduction in viability upon treatment with increasing doses of cisplatin compare to parental cell lines (Figure 1A)".  

Response: Changed as you indicated

  1. Line 166: referred figure is not correct: Figure 1B.

Response) Corrected.

  1.  Labels are not aligned (for example figure 3A, 4C "parental lines"; 5A middle WB; )

Response) Aligned.

Round 2

Reviewer 2 Report

Dear Authors,

Please correct following statements:

  1. Scale bar is missing or barely visible: Figure 2C; Figure 4A,B; Figure 7E; In the version that I am receiving no changes have been made regarding addition of visible scale bar.  There is no scale bar itself, only numbers.
  2. Figure 5. Font is not consistent with other figures.
  3. Please re-write the abstract (leave the only main message for article) for example lines 19-22 and remove "increased Comet tails and fluorescent p-H2AX for DNA damage response" it is misleading or change to: "increased levels of DNA damage (DSBs) evaluated by highly expressed pH2AX and DNA fragmentation registered by comet assay ..."
  4. The raising concerns regarding FACS data. H460 control and H460CR control plots on the bottom left contain late apoptotic events (subpopulation on the right upper quadrant - PI/Annexin positive events that are absent in treated condition). Please optimize the assay and provide solid data to convince the readers that apoptosis is seen at treated condition and early/late apoptotic events are more prominent at PGG rated cells.

Author Response

Reviewer 2

  1. Scale bar is missing or barely visible: Figure 2C; Figure 4A,B; Figure 7E; In the version that I am receiving no changes have been made regarding addition of visible scale bar.  There is no scale bar itself, only numbers.

(Response) Corrected for clear and visible scale bars.

  1. Figure 5. Font is not consistent with other figures.

(Response) Corrected as you indicated

  1. Please re-write the abstract (leave the only main message for article) for example lines 19-22 and remove "increased Comet tails and fluorescent p-H2AX for DNA damage response" it is misleading or change to: "increased levels of DNA damage (DSBs) evaluated by highly expressed pH2AX and DNA fragmentation registered by comet assay ..."

(Response) Corrected based on your comments

  1. The raising concerns regarding FACS data. H460 control and H460CR control plots on the bottom left contain late apoptotic events (subpopulation on the right upper quadrant - PI/Annexin positive events that are absent in treated condition). Please optimize the assay and provide solid data to convince the readers that apoptosis is seen at treated condition and early/late apoptotic events are more prominent at PGG rated cells.

(Response) Thanks. H460 control and H460CR control plots were replaced by new ones.

This manuscript is a resubmission of an earlier submission. The following is a list of the peer review reports and author responses from that submission.

Round 1

Reviewer 1 Report

Ji-Hyun Kim et al., has studied the “Apoptotic and DNA Damage Effect of 1,2,3,4,6-Penta-O-Gal-2 loyl-Beta-D-Glucose in cisplatin Resistant Non-Small Lung 3 Cancer Cells via Phosphorylation of γ-H2AX, CHK2 and p53. The authors have drastically modified the manuscript and many of the concerns has been addressed. However, improving the western blots, in particular the exposure of beta actin/AKT would enhance the data.

Author Response

We appreciate you for your kind advice regarding our exposure. Your comments will be very helpful for the upgrade of our future research.

Reviewer 2 Report

Comments to the author:

 The authors addressed the issue of in vitro and in vivo studies involving A549, H460 cells and BALB/c athymic nude mice to explore the apoptotic and DNA damaging effect of PGG in overcoming cisplatin resistance. Among others, their results showed that co-treatment of PGG and cisplatin in A549/CR and H460/CR cells synergistically increased the cytotoxicity, subG1 accumulation, enhanced cleavages of PARP, caspases and attenuated the expression of c-IAP and Bcl-2. The authors also showed that PGG suppressed the growth of H460/CR cells in Balb/c nude mice with increased expression of caspase 3 and concluded that PGG induces apoptosis through upregulation of DNA damage proteins and inhibition of anti-apoptotic proteins.

Although the topic of the manuscript is interesting, and the study is of sufficient significance and originality, there are serious shortcomings in the structure, content and presentation of data in the manuscript. Only a few of the many are listed:

  1. Abbreviations are not introduced in the abstract. Also, once introduced the abbreviations should be used through the manuscript.

  1. A lot of textual errors and typos.

e.g “Furthermore, Additionally, PGG suppressed the  20 growth of H460/CR cells in Balb/c athymic nude mice….” in row 20.

  1. A lot of inaccurate citations and use of references.

For example: in line 27 you said: “Lung cancers are known the most common malignancy in males worldwide [1].”

The cited reference [1] actually states: “lung cancer is the most commonly diagnosed cancer and the leading cause of cancer death in males, followed by prostate and colorectal cancer for incidence, and liver and stomach cancer for mortality.”

Also, in lines 151 – 152 you stated:

“It  is  well  known  that  multidrug  resistance  (MDR)  is  overexpressed and tumor suppressor PTEN is lowly expressed in resistant cancer cells compared to parental cells

[5].”, but neither MDR nor PTEN are mentioned in reference 5.

  1. The listed materials and methods are often incomplete and unclear.

For example: in lines 59 and 60 you said: “A549  and  H460  NSCLCs  from  American  Type  Culture  Collection   (ATCC)  were maintained in RPMI 1640 containing…”

but you did not specify the manufacturer of the media. It should be written: “A549  and  H460  NSCLCs  from  American  Type  Culture  Collection   (ATCC)  were maintained in RPMI 1640 (Gibco BRL, Grand Island, NY, USA) containing…”

  1. You did not list the concentrations of antibiotics (penicillin/streptomycin ) added to the medium (line 62)

  1. Cytotoxicity assay was written to be done for H460 and H460 / CR only (lines 63 – 70), but the results for A549 and A549 / CR are also shown (Fig.1)

  1. In the description of Cell Cycle Analysis (lines 71 – 76) no protocol for propidium iodide staining has been described. Attached reference 15 does not describe (doesn't even use) PI staining.

  1. Also, in the description of Cell Cycle Analysis (lines 71 – 76) you should specify how many cells per sample were analyzed.

  1. Repetitions of the same data, even in two consecutive sentences were observed.

For example: in lines 138 – 141:

“One week later, the mice were divided into three groups (n=4 per group); vehicle control 138 (1% DMSO with saline), PGG (2 mg/kg body weight) and cisplatin (2 mg/kg body weight) 139 groups. One week after implantation, the mice were orally administered with PGG or 140  given intraperitonial (i.p) injection with cisplatin daily for 12 days.”

  1. Legends of the figures are not written correctly. Some contain data that should be found in the material and methods (example 1), and some lack basic data (example 2).

Example 1: Figure 2. (lines 188 – 189)

“Cells were seeded at a density of 4 x 104 cells/ml and treated with PGG (25 μM) for 48 h. TUNEL staining was carried out and visualized under fluorescence microscopy (x60).”

Example 2: In the legend of figure 1a, the graphs for cell viability are not described. Also, it is not stated what are the controls (in relation to which the values are expressed).

  1. The discussion was written confusingly, with a lot of repetition and in bad English.

Based on all the above, my opinion is that this work without major corrections does not meet the criteria for publication in the “Cells”.

Author Response

The authors addressed the issue of in vitro and in vivo studies involving A549, H460 cells and BALB/c athymic nude mice to explore the apoptotic and DNA damaging effect of PGG in overcoming cisplatin resistance. Among others, their results showed that co-treatment of PGG and cisplatin in A549/CR and H460/CR cells synergistically increased the cytotoxicity, subG1 accumulation, enhanced cleavages of PARP, caspases and attenuated the expression of c-IAP and Bcl-2. The authors also showed that PGG suppressed the growth of H460/CR cells in Balb/c nude mice with increased expression of caspase 3 and concluded that PGG induces apoptosis through upregulation of DNA damage proteins and inhibition of anti-apoptotic proteins.

Although the topic of the manuscript is interesting, and the study is of sufficient significance and originality, there are serious shortcomings in the structure, content and presentation of data in the manuscript. Only a few of the many are listed:

  1. Abbreviations are not introduced in the abstract. Also, once introduced the abbreviations should be used through the manuscript.

(Response) Thanks. Added.

  1. A lot of textual errors and typos.

e.g “Furthermore, Additionally, PGG suppressed the  20 growth of H460/CR cells in Balb/c athymic nude mice….” in row 20.

(Response) Thanks. Corrected.

  1. A lot of inaccurate citations and use of references.

For example: in line 27 you said: “Lung cancers are known the most common malignancy in males worldwide [1].”

The cited reference [1] actually states: “lung cancer is the most commonly diagnosed cancer and the leading cause of cancer death in males, followed by prostate and colorectal cancer for incidence, and liver and stomach cancer for mortality.”

(Response) Thanks. Corrected.

  1. Also, in lines 151 – 152 you stated:

“It  is  well  known  that  multidrug  resistance  (MDR)  is  overexpressed and tumor suppressor PTEN is lowly expressed in resistant cancer cells compared to parental cells

[5].”, but neither MDR nor PTEN are mentioned in reference 5.

 (Response) Sorry for making you confused. Reference 5 was replaced by the suitable reference on MDR and PTEN.

5.The listed materials and methods are often incomplete and unclear.

For example: in lines 59 and 60 you said: “A549  and  H460  NSCLCs  from  American  Type  Culture  Collection   (ATCC)  were maintained in RPMI 1640 containing…”

but you did not specify the manufacturer of the media. It should be written: “A549  and  H460  NSCLCs  from  American  Type  Culture  Collection   (ATCC)  were maintained in RPMI 1640 (Gibco BRL, Grand Island, NY, USA) containing…”

 (Response) Thanks. The information on the manufacturer of media and reagents was added in this MS.

  1. You did not list the concentrations of antibiotics (penicillin/streptomycin ) added to the medium (line 62)

  (Response) Thanks. Added.

  1. Cytotoxicity assay was written to be done for H460 and H460 / CR only (lines 63 – 70), but the results for A549 and A549 / CR are also shown (Fig.1)

 (Response) Thanks. Added.

  1. In the description of Cell Cycle Analysis (lines 71 – 76) no protocol for propidium iodide staining has been described. Attached reference 15 does not describe (doesn't even use) PI staining.

 (Response) Thanks. Corrected by citing Reference 22.

  1. Also, in the description of Cell Cycle Analysis (lines 71 – 76) you should specify how many cells per sample were analyzed.

  (Response) Thanks. Added.

  1. Repetitions of the same data, even in two consecutive sentences were observed.

For example: in lines 138 – 141:

“One week later, the mice were divided into three groups (n=4 per group); vehicle control 138 (1% DMSO with saline), PGG (2 mg/kg body weight) and cisplatin (2 mg/kg body weight) 139 groups. One week after implantation, the mice were orally administered with PGG or 140  given intraperitonial (i.p) injection with cisplatin daily for 12 days.”

       (Response) Sorry for making you confused. Corrected..

  1. Legends of the figures are not written correctly. Some contain data that should be found in the material and methods (example 1), and some lack basic data (example 2).

Example 1: Figure 2. (lines 188 – 189)

“Cells were seeded at a density of 4 x 104 cells/ml and treated with PGG (25 μM) for 48 h. TUNEL staining was carried out and visualized under fluorescence microscopy (x60).”

Example 2: In the legend of figure 1a, the graphs for cell viability are not described. Also, it is not stated what are the controls (in relation to which the values are expressed).

        (Response) Thanks. The repeated information shown in Materials and methods was removed in Figure 2 based on your comments.

  1. The discussion was written confusingly, with a lot of repetition and in bad English.

         (Response) Thanks for your critical comments. We rewrote Discussion by removing repeated data or information.

Based on all the above, my opinion is that this work without major corrections does not meet the criteria for publication in the “Cells”.

Response) We appreciate you so much for your critical and organized comments. These will be very helpful for us to prepare our MS in the high quality journals including Cells in the future. 

Round 2

Reviewer 2 Report

I would like to thank the authors for addressing some of my initial comments. Unfortunately, the authors corrected only the examples given in the first report, without correcting similar errors in other parts of the manuscript.

  1. The first point of the first report indicated the incorrect entry of abbreviations.

The use of abbreviations was corrected in the abstract (which is given as an example of the required correction), but was not corrected in the rest of the text.

Example 1:

c-IAP1, c-IAP2 were mentioned three times (in the lines 107, 210, 223), but their full names were not mentioned at all.

Example 2:

RIPA buffer was mentioned in line 104, but its full name have not been mentioned in the text.

  1. Incorrect citations have not been corrected, including even this one from point 3 of the first report which reads:

A lot of inaccurate citations and use of references.

For example: in line 27 you said: “Lung cancers are known the most common malignancy in males worldwide [1].”

The cited reference [1] actually states: “lung cancer is the most commonly diagnosed cancer and the leading cause of cancer death in males, followed by prostate and colorectal cancer for incidence, and liver and stomach cancer for mortality.”

  1. Point 10. of the first report stated that legends of the figures are not written correctly, as they contain data that should be found in the material and methods.

Two examples given in the firs report were corrected, but the same errors in the rest of the legends remained uncorrected.

Figures 1. and 7., for example, still contain data that should be found in the material and methods section:

Figure 1: “Chemical structure of PGG. Molecular 177 Formula=C41H32O26. Molecular weight = 986 “ (lines 177, 178).

Figure 7: “Effect of PGG treatment H460/CR cells (2.5 × 106 cells in 100 µl 290 of Matrigel mixed with PBS) were subcutaneously injected in the right flank of BALB/c nude mice (6 weeks old) area.). Seven days later, mice (n = 4/group) were treated with vehicle (1% DMSO with saline), PGG (2 mg/kg body weight) by oral gavage or cisplatin (2 mg/kg body weight) by i.p. injection every day for 12 days.” (lines  290 – 293).

Data like this unnecessarily overload the text of the legend and should be found in the material and methods section.

  1. Point 4. of the first report stated: The listed materials and methods are often incomplete and unclear. Again, example were corrected, but the same errors in the rest of the legends remained uncorrected.

Example:

In the legend of the figure 1. and Materials and Methods, for Cytotoxicity Assay is written that cytotoxicity is determined, but cell viability is shown on the Figure 1?

These two terms (cytotoxicity and cell viability) are alternately used for the same parameter, without mentioning their relationship anywhere (should they be opposite phenomena?):

„Cisplatin did not exert cytotoxicity in A549/CR”( line 167)

“PGG significantly reduced the viability of A549/CR” (line 170)

  1. The discussion has been somewhat improved, there are fewer mistakes and it is easier to read, but still not clear enough, with confusing terminology and insufficiently corrected English.

In conclusion, I cannot recommend this paper for publication, as in my opinion it still suffers from numerous flaws and does not meet the minimum quality requirements necessary for publication in Cells.

Author Response

  1. The first point of the first report indicated the incorrect entry of abbreviations.

The use of abbreviations was corrected in the abstract (which is given as an example of the required correction), but was not corrected in the rest of the text.

Example 1:

c-IAP1, c-IAP2 were mentioned three times (in the lines 107, 210, 223), but their full names were not mentioned at all.

Example 2:

RIPA buffer was mentioned in line 104, but its full name have not been mentioned in the text.

(Response) Thanks Corrected 

  1. Incorrect citations have not been corrected, including even this one from point 3 of the first report which reads:

A lot of inaccurate citations and use of references.

For example: in line 27 you said: “Lung cancers are known the most common malignancy in males worldwide [1].”

The cited reference [1] actually states: “lung cancer is the most commonly diagnosed cancer and the leading cause of cancer death in males, followed by prostate and colorectal cancer for incidence, and liver and stomach cancer for mortality.”

(Response) Thanks. It was cited as what it is. 

“ lung cancer is the most commonly diagnosed cancer (11.6% of the total cases) and the leading cause of cancer death (18.4% of the total cancer deaths), closely followed by female breast cancer (11.6%), prostate cancer (7.1%), and colorectal cancer (6.1%) for incidence and colorectal cancer (9.2%), stomach cancer (8.2%), and liver cancer (8.2%) for mortality. Lung cancer is the most frequent cancer and the leading cause of cancer death among males, followed by prostate and colorectal cancer (for incidence) and liver and stomach cancer (for mortality)” 

  1. Point 10. of the first report stated that legends of the figures are not written correctly, as they contain data that should be found in the material and methods.

Two examples given in the firs report were corrected, but the same errors in the rest of the legends remained uncorrected.

Figures 1. and 7., for example, still contain data that should be found in the material and methods section:

Figure 1: “Chemical structure of PGG. Molecular 177 Formula=C41H32O26. Molecular weight = 986 “ (lines 177, 178).

Figure 7: “Effect of PGG treatment H460/CR cells (2.5 × 106 cells in 100 µl 290 of Matrigel mixed with PBS) were subcutaneously injected in the right flank of BALB/c nude mice (6 weeks old) area.). Seven days later, mice (n = 4/group) were treated with vehicle (1% DMSO with saline), PGG (2 mg/kg body weight) by oral gavage or cisplatin (2 mg/kg body weight) by i.p. injection every day for 12 days.” (lines  290 – 293).

Data like this unnecessarily overload the text of the legend and should be found in the material and methods section.

(Response) Thanks Corrected. Also, Figure1B chemical structure was removed, since it is well known through Google. 

  1. Point 4. of the first report stated: The listed materials and methods are often incomplete and unclear. Again, example were corrected, but the same errors in the rest of the legends remained uncorrected.

Example:

In the legend of the figure 1. and Materials and Methods, for Cytotoxicity Assay is written that cytotoxicity is determined, but cell viability is shown on the Figure 1?

These two terms (cytotoxicity and cell viability) are alternately used for the same parameter, without mentioning their relationship anywhere (should they be opposite phenomena?):

„Cisplatin did not exert cytotoxicity in A549/CR”( line 167)

“PGG significantly reduced the viability of A549/CR” (line 170)

 (Response) Thanks Corrected 

  1. The discussion has been somewhat improved, there are fewer mistakes and it is easier to read, but still not clear enough, with confusing terminology and insufficiently corrected English.

  (Response) Thanks. Corrected 

In conclusion, I cannot recommend this paper for publication, as in my opinion it still suffers from numerous flaws and does not meet the minimum quality requirements necessary for publication in Cells.

(Response) We appreciate a respectful reviewer for giving us valuable comments. These comments will be very helpful to us for high quality journal publication in the future.